# Clickable Biomaterials for Modulating Neuroinflammation

**DOI:** 10.3390/ijms23158496

**Published:** 2022-07-31

**Authors:** Chase Cornelison, Sherly Fadel

**Affiliations:** Department of Biomedical Engineering, University of Massachusetts Amherst, Amherst, MA 01003, USA; sfmakar@umass.edu

**Keywords:** biomaterial, scaffold, particles, bioorthogonal, click chemistry, immunomodulation, neural injury, neurodegeneration, neuroinflammation

## Abstract

Crosstalk between the nervous and immune systems in the context of trauma or disease can lead to a state of neuroinflammation or excessive recruitment and activation of peripheral and central immune cells. Neuroinflammation is an underlying and contributing factor to myriad neuropathologies including neurodegenerative diseases like Alzheimer’s disease and Parkinson’s disease; autoimmune diseases like multiple sclerosis; peripheral and central nervous system infections; and ischemic and traumatic neural injuries. Therapeutic modulation of immune cell function is an emerging strategy to quell neuroinflammation and promote tissue homeostasis and/or repair. One such branch of ‘immunomodulation’ leverages the versatility of biomaterials to regulate immune cell phenotypes through direct cell-material interactions or targeted release of therapeutic payloads. In this regard, a growing trend in biomaterial science is the functionalization of materials using chemistries that do not interfere with biological processes, so-called ‘click’ or bioorthogonal reactions. Bioorthogonal chemistries such as Michael-type additions, thiol-ene reactions, and Diels-Alder reactions are highly specific and can be used in the presence of live cells for material crosslinking, decoration, protein or cell targeting, and spatiotemporal modification. Hence, click-based biomaterials can be highly bioactive and instruct a variety of cellular functions, even within the context of neuroinflammation. This manuscript will review recent advances in the application of click-based biomaterials for treating neuroinflammation and promoting neural tissue repair.

## 1. Introduction

Neuroinflammation is an inflammatory state generally considered to be specific to the central nervous system (CNS). While communication between neural and immune cells can be beneficial, such as during tissue development and synapse maintenance [1], the connotation of ‘neuroinflammation’ is detrimental immune cell involvement in CNS tissue that induces neurotoxicity and/or neural dysfunction. Events like traumatic injury, neurodegenerative disease, neurotoxin exposure, local or systemic infection, metabolic syndromes like diabetes and obesity, and even natural aging can induce a state of neuroinflammation [2,3,4,5]. The resulting pathological effects are mediated and propagated by CNS-resident agents of the immune system, such as microglia and astrocytes, as well as those from the periphery, like circulating monocytes and leukocytes [3]. These cells undergo spatiotemporal changes in morphology, proliferation rate, and expression of inflammatory cytokines and markers of activation, which drive neuropathological outcomes [2,6]. Advancing our understanding of the dynamics of neuroinflammation and how to modulate or redirect immune-CNS interactions will be critical for developing therapeutic strategies for neural tissue stabilization or regeneration. 

The manipulation of the immune system for therapeutic purposes is an emerging field known as immunomodulation or immunotherapy. In general, therapeutic molecules are delivered into the body to instruct or redirect the functions of immune cells to promote natural repair processes. Immunomodulation strategies have already provided promising clinical results for the treatment of recalcitrant tumors [7,8] as well as some autoimmune diseases [9]. The potential for off-target effects or dosage limitations with systemic delivery has motivated the use of localized delivery as a preferred route of administration [10]. Toward this goal, injectable or implantable biomaterials offer several advantages over free-drug formulations, namely controlled, sustained release; increased cell or tissue targeting; and enhanced payload delivery [10,11,12]. Historically, implantable biomaterials were bioinert to avoid mounting a foreign body response by the host immune system. More contemporary biomaterial strategies have sought to actively engage the immune system, becoming a powerful tool for augmenting tissue integration and repair [10,13]. Materials based on so-called “click” chemistry are of particular interest because click reactions are bioorthogonal and proceed in the presence of biological molecules or within living systems without interfering with native biochemistry [14,15]. 

Bioorthogonal chemistries were initially applied to fundamental studies of biomolecules in live cells but have been rapidly expanded for biomaterial polymerization, crosslinking, and functional decoration [16,17,18]. While several such bioorthogonal biomaterials have been applied toward treating neuroinflammation or promoting neural tissue regeneration, it can be difficult to identify many of the references due to variations in diction or approach descriptions. Hence, our goal is to aggregate resources describing the implementation of bioorthogonal biomaterials for modulating neuroinflammation, prior to which we review the contexts and challenges of treating neuroinflammation and the therapeutic potential of using biomaterials for immunotherapy. 

## 2. Effectors, Stimuli, and Outcomes of Neuroinflammation

Neuroinflammation is driven by the expression of pro-inflammatory cytokines and reactive oxygen species which collectively act to recruit immune cells to the site of pathology; stimulate proliferation, inflammatory activation, and debris clearance; and potentially induce unwanted tissue damage [3]. While inflammation has a negative connotation, immune cell involvement is both necessary and beneficial for early tissue protection in the CNS. In fact, inhibiting immune cell involvement early after CNS injury exacerbates tissue damage [19,20,21,22,23,24]. Nonetheless, sustained neuroinflammation at chronic stages also inhibits tissue repair [2,25,26]. This inflammatory environment is potentiated by both resident glial cells and extravasated peripheral immune cells. The primary glial cells known as astrocytes and microglia both directly contribute to the inflammatory milieu and modulate the response of infiltrating peripheral immune cells, namely monocytes/macrophages, T cell, mast cells, and neutrophils [24,27]. 

### 2.1. Cell Types and Phenotypes in Neuroinflammation

Astrocytes are the most common glial cell in the CNS and perform a wide range of functions, including maintenance of the blood-brain and blood-spinal cord barriers, transport of nutrients and neurotransmitters to neurons, and communication with other neural cells. Microglia are the tissue-resident macrophages of the CNS and therefore act as the first line of defense against neural trauma, infection, or disease. Peripheral immune cells are also often recruited to the CNS by the release of chemotactic factors and the breakdown of the blood–brain barrier (BBB) following an injury or onset of disease [28]. Both macrophages and microglia are derived from a monocyte lineage and have long been known to exhibit a spectrum of phenotypes ranging from neuroprotective to neurotoxic [2,29]. Recently, astrocytes were also shown to exist on a similar spectrum of phenotypes [30,31,32]. At end of the spectrum lies pro-inflammatory or classical activation, denoted as M1 and A1 for microglia/macrophages and astrocytes, respectively. The converse is anti-inflammatory or alternative activation, designated as M2 and A2, respectively. While this notation implies a dichotomy, the same cell can simultaneously express markers for both pro-and anti-inflammatory phenotypes. Hence, more research is needed to better characterize how cells adopt intermediate phenotypes and the functional impacts of such. 

Pro-inflammatory (M1 and A1) phenotypes are activated mainly by pathogens or pro-inflammatory stimuli, such as bacterial lipopolysaccharide (LPS), tumor necrosis factor-alpha (TNFα), and interferon-gamma (IFNγ) [24,33]. For microglia/macrophages, the M1 phenotype is characterized by activation of the signal transducer and activator of transcription 1 (STAT1) pathway and expression of pro-inflammatory cytokines such as TNFα; interleukins IL-16, IL-1β, IL-18; and reactive oxygen and nitrogen species [2,34,35]. The M1 phenotype is often identified by the expression of membrane receptors CD16, CD32, and CD80, and the chemokines CCL5, CCL20, CXCL1, and CXCL9 [35]. The neurotoxic A1 phenotype is associated with pro-inflammatory pathways such as mitogen-activated protein kinase (MAPK) and nuclear factor kappa b (NF-κB) and is induced by signaling from pro-inflammatory microglia via IL-1α, TNF-α, and C1q [30,32,36,37]. A1 astrocytes express soluble factors like complement C3, IL-1β, TNF-α, and nitric oxide, and genes such as Serping1, Srgn, and Amigo2 [30]. It was also shown that A1 astrocytes promote neurotoxicity via saturated lipids. Other pro-inflammatory cytokines include GM-CSF, IL-6, and several CXCLs and CCLs. Unfortunately, both astrocytes and microglia display characteristics of pro-inflammatory activation with normal aging [38,39,40], suggesting M1/A1 phenotypes may contribute to age-related cognitive decline. Inflammatory mediators, such as LPS, therefore can act directly on microglia and still indirectly stimulate astrocyte activation [41].

In contrast, the M2 phenotype is activated by anti-inflammatory pathways such as STAT6 via anti-inflammatory cytokines: IL-4, and IL-13, or by activation of STAT3 via the expression of IL-10 [2]. Activation of the anti-inflammatory M2 phenotype is generally characterized by the expression of markers such as CD206, ARG1, IL-10, IL-4, IL-13, and transforming growth factor-ꞵ (TGFꞵ) [2,35]. A2 astrocytes may be regulated by the Janus kinase/STAT3 pathways and produce IL-4, IL-10, and TGF-β following exposure to the anti-inflammatory cytokines, IL-4, IL-13, and IL-10. These protective pathways can be activated by glycoprotein gp130, a signal transducer for the IL-6 cytokine, as well as cytokines such as TGFβ and IFN-γ. Astrocytes also coordinate with neurons for A2 activation, wherein neuronal EphB1 promotes A2 astrocyte phenotypes associated with BDNF secretion. This neuronal-activated pathway was found to activate neuronal STAT3, ultimately preventing neuronal loss and demyelination and disrupting glial scarring [2,31]. 

### 2.2. Neural Pathologies and the Role of Glial Cells 

Inflammation is an underlying condition of neural responses to trauma, autoimmune diseases, and neurodegenerative diseases such as multiple sclerosis, Alzheimer’s disease, and Parkinson’s disease [2,6]. There are shared characteristics across these diverse pathologies, but each type can also have disease-specific factors or functional consequences. Alzheimer’s disease is associated with intracellular tau accumulation and extracellular accumulation of Aβ protein, which induce microglial and astrocyte activation and secretion of pro-inflammatory cytokines [42]. The inflammatory stimulus in Parkinson’s disease is related to intracellular accumulation of Lewy bodies clumps and α-synuclein protein [43]. Multiple sclerosis is characterized by spinal infiltration of myelin-autoreactive T cells which seek and destroy the myelin sheath [44]. In this context, astrocytes, microglia, and even neurons also play direct and indirect roles in regulating immune cell phenotypes and functions [2,27,45,46]. The disease-specific contexts are therefore important to consider when designing a strategy to treat or overcome neuroinflammation. 

The CNS exhibits a unique response to traumatic injury. Astrocytes respond to trauma by enhanced proliferation and higher expression of proteins like glial fibrillary acidic protein (GFAP), vimentin, nestin, and synemin than when resting [32,35]. While these cells are not traditionally considered immune cells, astrocytes do function as phagocytes and antigen-presenting cells [47,48]. Infiltrating macrophages and resident microglia acutely exhibit an anti-inflammatory phase that transitions into a prolonged, persistent pro-inflammatory phase [49,50]. This response is contrary to that of peripheral tissues and is an active barrier to recovery [51]. One contributing factor to such a contradictory response in the CNS is the so-called glial scar. In the acute phase of injury, the glial scar protects adjacent healthy tissue and limits injury expansion; but the scar also physically and chemically inhibits neuronal regeneration at chronic times [52,53]. Additionally, the matrix components of the scar, such as collagen IV, laminin, and chondroitin sulfate proteoglycans, actively stimulate pro-inflammatory activation of infiltrating immune cells [54,55]. Nonetheless, depletion of scar-producing astrocytes five weeks after injury leads to lesion expansion and additional tissue degeneration [22]. A2 astrocytes are indeed present two weeks after injury, and expression of A2 genes positively correlates with neurological recovery [56]. Microglia are also vital for neuroprotection, in part by actively promoting proliferation and activation of scar-forming astrocytes [19,20]. More research is needed to understand how glial scarring relates to astrocyte phenotypes and if it is possible to promote eventual scar resolution. 

The above data highlight the complex orchestra that is the post-injury neural tissue environment and the nuance requirements for therapeutically combating neuroinflammation. It is important to consider temporal and disease-specific contexts of both the immune and glial cell compartments when attempting to modulate neuroinflammation, and therapeutic strategies to control cellular phenotypes will advance the ability to target or redirect detrimental inflammatory responses. Future strategies to combat neuroinflammation will benefit from the use of biomaterials given their versatility and effectiveness as either passive vectors or active agents for local drug delivery and immune cell modulation.

### 2.3. Role of Peripheral Immune Cells

Similar to microglia and astrocytes, the functions of macrophages depend on the surrounding environment, which can depend on the type and stage of injury or disease [57]. In many contexts, the inflamed CNS experiences an ingress of bone marrow-derived monocytes, which differentiate into monocyte-derived macrophages and play a pleiotropic role in neuroinflammation [58]. Macrophage interactions with myelin debris and chondroitin sulfate proteoglycans in the glial scar induce the production of pro-inflammatory cytokines and reactive species that propagate neurotoxic inflammation [59,60]. Conversely, the presence of bone marrow-derived macrophages also mitigates microglial reactivity and ultimately reduces inflammation after CNS injury [61,62]. Studies have also shown that inhibiting macrophage migration disturbs glial scar formation and leads to enhanced neuronal and functional losses [52]. Administration of IL-4 and IL-10 after injury facilitates neuronal protection, myelination, and functional recovery [63]. Following the injury, activated macrophages and microglia must undergo apoptosis or they will likely persist into the chronic phase [58,64]. 

T cells are another important cellular element in neuroinflammation [65,66]. Naïve T cells surveil the CNS from within the dural sinuses of the cerebral and spinal meninges and also traffic within the meningeal lymphatic vessels [67]. In fact, meningeal lymphatics play a major role in regulating CNS immunity in health and disease [68,69,70]. T cell egress from the meningeal lymphatics is mediated by the C-C chemokine receptor type 7 (CCR7), expression of which decreases with age and is linked to an increase in detrimental T cell subsets [71]. This age-associated decrease in T cell trafficking is also directly linked to cognitive impairment. Once a T cell is activated, it becomes a cytotoxic CD8+ cell or a helper CD4+ cell, which further differentiates into different subsets: Th1, Th2, Th17, Treg (regulatory), and Tfh (follicular helper) [65]. 

Each T cell subset has a specific phenotype, function, and role in inflammation, with either a pro- or anti-inflammatory effect. For example, Th1 cells are triggered by IL-12 signaling, STAT4, and STAT1, and produce IFN-γ, IL-6, IL-10, TNF-β, and IL-2. The resulting pro-inflammatory state is therefore denoted as type 1 immunity (i.e., associated with Th1 cells). The Th2 phenotype is induced by the GATA-3 pathway, which is enhanced by NF-kB signaling and results in inhibitory cytokines such as IL-4, IL-13, IL-31, and IL-5 that trigger a type 2 inflammatory response. Treg cells tend to exert immunosuppressive effects, which can be beneficial in cases of trauma [72] but detrimental in cancer [73]. Lastly, despite the ‘helper’ moniker, Th17 and Tfh cells are implicated in autoimmune disorders such as multiple sclerosis and directly damage the myelin sheath and oligodendrocytes [74,75]. While it is possible for tissue-resident microglia to engulf living, pathogenic T cells [76], this ability is likely overwhelmed at high rates of T cell infiltration. Other immune cells with an active role in neuroinflammation include neutrophils [77,78,79], mast cells [80], and antibody-producing B cells [81]. 

## 3. Biomaterial Strategies to Modulate Immune Cell Phenotypes

Regulating or manipulating the immune response is a promising strategy to halt or treat neurological damage. Early iterations of immunomodulation, or immunotherapy, used systemic delivery of small molecules or protein therapeutics [10]. The most notable example may be PD1 inhibitors in cancer, which prevent the induction of T cell apoptosis and immunosuppression by cancer cells [82,83]. Specific to the CNS, the FDA approved the small molecule fingolimod for the treatment of multiple sclerosis in 2018. The drug fingolimod prevents T cell infiltration into the spinal cord by blocking sphingosine 1-phosphate (S1P) signaling [84]. Fingolimod also alters the function of astrocytes, pericytes, and oligodendrocytes [85,86,87,88,89], suggesting systemic delivery may have secondary or off-target effects. Localized or triggered release of such a therapeutic cargo can be accomplished using biomaterials. Biomaterials have been used for prophylactic vaccines [90], cancer therapy [91,92], treating autoimmune diseases like diabetes [93], and promoting tissue regeneration [94,95,96]. We briefly review a few notable examples, but further review of CNS-specific or biomaterial-based immunomodulation is available [10,18,97,98,99,100,101,102,103].

Implanting a biomaterial inherently stimulates interactions with the host immune system, dictated in part by the preceding cascade of protein adsorption and provisional matrix formation. These interactions at the cell-biomaterial interface provide an opportunity to steer the resulting immune response. Biomaterials can be formulated as particles, scaffolds, or cross-linkable hydrogels [102,104,105,106]. Engineering the characteristics of a material and/or encapsulating drug cargo can enable control over cellular recruitment and phenotypic biasing, including immune cells. Additionally, conjugation or encapsulation of active molecules like peptides and proteins protects this fragile cargo from premature clearance or degradation by harsh physiological conditions [107,108]. Of course, it is critically important to choose an appropriate biomaterial and drug cargo to achieve a desired stromal, stem cell, or immune cell outcome [10,109,110,111]. Therapies for immunomodulation often leverage known pro- or anti-inflammatory pathways for cancer therapy and regenerative medicine, respectively. For example, anti-cancer therapies may deliver IL-1β, TNF-α, or agonists of Toll-like receptors, while anti-inflammatory approaches may include TGF-b, IFNγ, or interleukins IL-4, IL-6, IL-10, and IL-13. One consideration is that anti-inflammatory approaches may also benefit from small doses of pro-inflammatory signals—at least initially—since M2 macrophages derived from M1 cells are more effective at promoting angiogenesis than M2 cells derived from unstimulated M0 cells [112]. Figure 1 summarizes how materials can be engineered to impact neuroinflammation. 

### 3.1. Particulate Biomaterials

Nano- and microparticle formulations leverage the ability of many immune cells to engulf or phagocytose particulate materials loaded with therapeutic drugs. Biomaterial particles are therefore an effective strategy to modulate immune cell phenotype from within the cell itself [113]. Phagocytosis is influenced by the size, shape, charge, stiffness, and fluidity of the biomaterial particle [107]. Round particles are more readily phagocytosed, but rod-like particles induce a larger immune response [114]. Rod-shaped particles also enhance the inflammatory response against decorated ligands, regardless of the ligand-receptor targets [115]. Particles smaller than 3 microns are good for intracellular drug delivery, while particles larger than 10 microns are better for extracellular delivery [114,116]. Combining multiple particle sizes can enable a multifaceted approach [116], as can co-loading multiple stimulating factors into the same particle. For example, co-loading agonists of the stimulator of interferon genes (STING) pathway and Toll-like receptor 4 yields a stronger type I IFNβ immune response for eliminating cancer cells [92]. 

In contrast to inducing phagocytosis, coating particles with poly(ethylene glycol) (PEG) will protect them from immune recognition and lengthen blood circulation time. Nonetheless, PEGylated particles must strike a balance between immune evasion, avoiding off-site accumulation, and desirable uptake by target immune cells, promoting the desired effect [117]. Nanoparticles can also be attached to living cells as a “cellular backpack”, which enhances particle homing and accumulation at target sites. Cellular backpacks are designed to stick to trafficking cells without being phagocytosed or undesirably altering cell function. Importantly, immune cells with backpacks can still cross the BBB [118]. Backpack-laden macrophages address some disadvantages of conventional carriers by enabling less cargo to be loaded, reaching the target site during inflammation, and protecting and sustaining the release of the drug cargo [118,119,120].

One common polymer used for particle development is poly lactic-co-glycolic acid or PLGA. This copolymer has been used to deliver an array of immunomodulatory agents to redirect immune cell phenotype and function, including TGFꞵ [116,121], granulocyte-macrophage colony-stimulating factor (GM-CSF) [116,122]; agonists of the STING pathway [123]; steroid hormone vitamin D3 [122]; vaccine adjuvant [124]; and the antioxidant N-acetylcysteine [125]. The PLGA polymer itself may be immunosuppressive as it degrades into lactic acid, which causes resistance to pro-inflammatory stimulation by LPS [126]. Other engineered particle formulations include those based on lipids [92], gold [127,128], and metal oxides [129], to name a few.

Cells naturally produce lipid-based nanoparticles called extracellular vesicles (EVs), which enable long-distance cell–cell communication. The most widely recognized type of EV is exosomes, which are emerging as powerful tools to assess tissue damage, deliver therapeutic molecules, and directly or indirectly modulate immune responses [130,131,132]. EVs play a key role in immunomodulation and have been shown to be the mechanism through which implanted cells like mesenchymal stem cells promote immunosuppression and wound resolution [133,134]. One key advantage to exosomes for neural applications is their innate ability to cross the BBB, an ability which is further enhanced under inflammatory conditions [135]. EVs can therefore leverage the conditions of neuroinflammation as a driving force for targeted drug delivery in the CNS. 

EVs contain surface receptors and intracellular cargo derived from the source cell and can be engineered using genetic manipulation or exogenous chemistries for enhanced drug delivery or cellular targeting [136,137,138,139]. For further compositional control, engineered liposomes can be developed in vitro using sonication, forced filtration, or microfluidic methods [140]. A user controls what lipids are contained with the bilayer, the contents of the liposome interior or surface, which route through which the particles will be taken up, and the ability to target specific cell types. While standalone EVs and liposomes have many advantages, loading the particles into scaffolds can deter unintended phagocytosis and enable controlled or sustained release, and preferential accumulation at the target site [141,142]. 

### 3.2. Solid Scaffolds and Hydrogels

Biomaterial scaffolds also have great prospects for immunomodulation, as reviewed by [102,143]. Physicochemical properties of the scaffold direct immune cell adhesion, morphology, proliferation, and polarization and can be tuned by altering material surface chemistry or topography; mechanical properties; geometry, shape, and pore size; and degradation rates or byproducts [96,144]. Materials with properties similar to native tissue tend to promote more anti-inflammatory or pro-regenerative immune cell outcomes. For example, scaffolds comprising polymer fibers promote more M2 macrophages, potentially as a result of mimicking the fibrous qualities of collagen-based tissues [145,146]. Fibrous scaffolds are also inherently porous, and both fiber size and pore size are positively correlated with increased expression of M2 markers [145,147]. Related to mechanical properties, materials with Young’s moduli on the order of most native tissues (1–100 kPa) also promote an anti-inflammatory phenotype, while stiffer matrices prime the cells toward a pro-inflammatory phenotype [148]. Nonetheless, this macrophage response can depend on the crosslinking agent used to modulate the material stiffness, since genipin crosslinking appears to suppress all immune responses while crosslinking with carbodiimide enhances pro-inflammatory responses, specifically [149,150,151]. 

The chemical properties of a material, such as bulk and surface composition, surface charge and density, and hydrophilicity/phobicity influence protein adsorption and therefore immune cell interactions [103]. Again, a distinct advantage may be afforded to materials chemically similar to—or derived from—native tissue, since these materials mimic the physiological environment. Native tissue can be rendered non-immunogenic through decellularization—the process of treating tissue with detergents, enzymes, and/or other reagents to remove cellular components and isolate the extracellular matrix (ECM). Tissue-derived ECM is a natural source of instructive cues for many cellular processes and has shown vast potential as an immunomodulatory biomaterial [152,153]. Decellularized tissues induce a T helper 2 cell response that guides IL-4–dependent (anti-inflammatory) macrophage polarization [95,154]. Complete removal of cellular debris is required, however, since incomplete decellularization promotes a pro-inflammatory response [155]. 

ECM-driven induction of anti-inflammatory phenotypes has been ascribed to the presence of residual hyaluronan [156] as well as matrix-bound nanovesicles [157,158]. Interestingly, the phenotype induced by exposure to ECM scaffolds appears functionally distinct from canonical anti-inflammatory phenotypes, suggesting the response is more nuanced and requires additional study [157]. Nonetheless, one aspect is clear: decellularized ECM in its native, structured form or as an injectable hydrogel is an effective scaffold for immunomodulation and tissue repair, including in the CNS [159,160,161,162]. 

### 3.3. Materials for Glial Cell Modulation

Material implantation into the CNS will recruit and bias the phenotype of not only peripheral immune cells but also resident glial cells. Intentionally designing materials to positively interact with, and dictate the phenotype of, glial cells may be an impactful approach to sustained immunomodulation since these cells play direct roles in regulating and propagating inflammatory responses. As a baseline, nonionic and anionic materials appear to be best for avoiding a detrimental foreign body response in vivo [163]. In vitro studies have informed much of what is known about astrocyte responses to biomaterials [164,165,166,167], but the recent description of A1/A2 astrocyte phenotypes means much more work is necessary to understand how to engineer materials that intentionally harness astrocyte for repair. In mechanically dynamic materials, matrix stiffening limits reactive astrogliosis while softening of the material enhances astrocyte activation in 3D [168]. These mechanical effects may be related to the astrocytic scar, which was once assumed to be stiff but more recently described as being softer than healthy CNS tissue [55]. Fibrous and porous materials enable astrocyte invasion into the material, and aligned fibers promote bipolar or stellate morphologies common to quiescent astrocytes. Cultured astrocytes on larger fiber diameters appear to support more neurite outgrowth compared to smaller fiber diameters [169]. Nonetheless, electrospun fibers of PLLA were found to the enhance expression of A1, pro-inflammatory genes, requiring controlled release of TGF-β to promote neuroprotective A2 phenotypes [167]. For further review of how engineered biomaterials modulate glial responses and neuroinflammation, see the recent review by Tsui et al. [98]. 

Next-generation therapies will leverage temporal and spatial control over bulk or local material properties, ligand presentation, and drug release profiles. Enacting control over one or more of these axes can be difficult in vivo, but leveraging chemistries that do not interact with biology—known as bioorthogonal reactions—can enable spatial and temporal alterations in material composition, even after implantation. This versatility can enable therapeutics to combine dynamic mechanical or chemical changes, incorporation of other materials such as liposomes or exosomes, and/or triggered the release, exchange, or refilling of multiple drug cargos. In the next section, we introduce a range of bioorthogonal chemistries currently used and discuss their applications in biomaterials science. 

## 4. Bioorthogonal ‘Click’ Chemistry and Neural Biomaterials Applications

Click reactions are generally ligation reactions conducted under ambient conditions which rapidly and selectively produce the desired ligation product in high yield [14,170]. Several click reactions have been developed to date, enabling an array of chemical ligation strategies for bioorthogonal polymerization, crosslinking, or conjugation [14,17]. The earliest and most widely used click reaction in polymer science is the copper-catalyzed azide-alkyne cycloaddition (CuAAC). Given concerns about copper toxicity, numerous copper-free click reactions have also been developed and seen wider adoption in the biomaterials field. These reactions are used for tuning crosslinking kinetics and dynamics [171,172,173]; directing site-specific conjugation [174,175,176,177]; polymerizing and functionalizing hydrogel microparticles [178]; controlling small molecule loading and reloading [179], and investigating cellular mechanobiology [180]. Typically, the reaction scheme involves simply mixing polymers and biomolecules containing the click reactive groups. Endogenous proteins or glycans can also be labeled through the cellular metabolism of non-canonical amino acids or unnatural sugar residues, making it possible to engineer otherwise less-controllable materials like cell-derived matrices or decellularized tissues [175,181]. Below we highlight the most common copper-free click chemistries and highlight key biomaterial applications. Figure 2 depicts the most common reaction schemes discussed here. For further review of click reactions in biomaterials, refer to Anseth and Klok [14]; Madl and Heilshorn [170]; Arkenberg, Nguyen, and Lin [171]; and Idiago-López et al. [182]. 

### 4.1. Copper-Free Strain-Promoted Azide-Alkyne Cycloaddition (SPAAC)

Since the initial discovery by Bertozzi and colleagues [183], strain-promoted azide-alkyne cycloaddition (SPAAC) has proven to be an effective alternative to CuAAC, maintaining the high specificity and reaction rate while limiting cytotoxicity. Known SPAAC reactions commonly used in biomaterials include the 1,3-dipolar cycloaddition between azides (a 1,3-dipole, -N_3_) and cyclooctynes, like dibenzocyclooctyne (DBCO) [184,185,186]; the tetrazine ligation [187,188]; and oxime chemistry with aldehydes and ketones [189]. Other reactions include the isocyanide-based click reaction [190] and the quadricyclane ligation [191]. 

SPAAC reactions can be used as the primary mechanism of hydrogel crosslinking for gentle and efficient cell encapsulation. For example, Hardy, Lin, and Schmidt used oxime click chemistry to crosslink linear, homo-bifunctional aminooxy PEG with aldehyde-functionalized hyaluronic acid (HA) [189]. Additionally, engineered elastin-like proteins have been developed to crosslink via SPAAC, which gelled completely within minutes and supported encapsulation and maintenance of murine neural progenitor cells. The stiffness and cell adhesion properties of the SPAAC-ELP gels could be tuned independently through the incorporation of RGD peptides. The same group later modified the elastin-like protein to contain azide-functionalized tyrosine residues for SPAAC to overcome the reliance on amine-reactive chemistry [192]. 

SPAAC reactions are commonly used to functionalize hydrogel biomaterials with molecules of interest. Azide-bearing nerve growth factor has been selectively grafted onto an acrylate-terminated, poly(β-amino ester) form of PEG-diacrylate [193]. The resulting hydrogels were then subsequently conjugated with thiol-containing CYIGSR adhesive peptides through a thio-ene reaction (discussed below) to further promote neuroregeneration. Adil et al. used a SPAAC reaction to first incorporate azide-containing RGD peptide into DBCO-functionalized HA, then used the remaining unreacted DBCO groups to crosslink the hydrogel with homobifunctional PEG-azide [186]. This approach yielded a 5.6-fold improvement in transplanted cell survival compared to 2D cultured cells and a 5.4-fold increase in dopaminergic neuron differentiation and survival at 4 months post-injection. Similarly, Li and colleagues decorated DBCO-functionalized PEG with azide-modified laminin and interferon proteins prior to crosslinking with four-arm PEG-tetraazide [194]. This approach enabled in vitro culture of neural stem cells over a two-week period and specified for primarily neuronal differentiation without supplementing differentiation factors in the medium.

Leveraging the bioorthogonal nature of these reactions can also provide temporal control over biomolecule presentation. Micropatterned brushes of PEG-azide have been modified after the initiation of cell culture simply by adding peptide–DBCO conjugates into the media [184]. Additionally, azide-modified alginate hydrogels have been filled and refilled long after in vivo implantation using intravenous delivery of cyclooctyne-conjugated small molecules [179]. One can envision using this approach to alter drug dosage or completely change to a new drug based on the patient’s therapeutic response. Hence, temporal control over drug-loaded biomaterials is a step in the right direction for personalized therapy. 

### 4.2. Diels-Alder and Inverse Electron-Demand Diels-Alder

Diels-Alder reactions occur between a diene (a molecule with two adjacent double bonds) and a substituted alkene (one double bond), which is commonly called the dienophile [195]. These are cycloadditions promoted not by strain but by electron rearrangement, either by normal or inverse electron demand. Normal electron-demand reactions occur between electron-poor dienophiles and electron-rich dienes, while inverse electron-demand reactions occur between an electron-rich dienophile and an electron-poor diene. Common diene-dienophile pairs used in biomaterials are maleimide-furan and tetrazine-norbornene for normal and inverse electron-demand Diels-Alder reactions (IEDDA), respectively. For a more in-depth review of these reactions and applications in biomaterials, see [196,197].

Furan modified-HA is a common Diels-Alder polymer backbone used for neural applications, which is typically crosslinked using PEG-bismaleimide. The mechanical and degradation properties of the resulting Diels−Alder cross-linked hydrogels can be tuned by varying the molar ratio of furan to maleimide [172]. These hydrogels can be used to encapsulate and deliver cells in vivo, such as V2a interneurons derived from mouse embryonic stem cells [198]. Another application is the production of cryogels, in which the uncrosslinked polymer mixture is frozen to induce ice crystal formation and then thawed to induce crosslinking and preservation of the porous architecture [199]. After crosslinking, maleimide-functionalized peptides can still be clicked into the gel via remaining unreacted furan groups to add cell adhesion sites or other bioactive motifs. Nonetheless, crosslinking between furan and maleimide can be relatively slow, on the order of hours. To improve the gelation rate, Madl and Heilshorn replaced the furan group with fulvene, which crosslinked with maleimide an order of magnitude faster than other common Diels-Alder reaction pairs while remaining stable for months [200]. Yet, another caveat is that many Diels-Alder reactions are most efficient at acidic pH, which is not optimal for cell encapsulation. Smith et al. addressed this concern by replacing the furan group on hyaluronic acid with the more electron-rich methylfuran to achieve faster, more efficient gelation at physiological pH [201]. 

IEDDA reactions can also be used as the primary mode of hydrogel gelation for cell encapsulation. Hydrogel properties like gelation rate and stiffness can be adjusted by varying the degree of polymer modification and molar ratio of tetrazine to norbornene. Stiffer, faster gelling materials are achieved at higher percent functionalization and an equimolar ratio of reactive groups [202]. Notable applications of IEDDA crosslinked biomaterials include assembling microporous annealed particle hydrogels for injection after stroke [203,204]; enabling the formation of low polymer content (and therefore low viscosity) methylcellulose for co-delivery of neural progenitor cells and scar-degrading enzyme [205]; and demonstrating maintenance of neural stemness depends on hydrogel degradation and cell–cell contacts regardless of stiffness [192]. 

### 4.3. Photo-Click Thiol-Ene Reactions

Photo-induced chemistries can be used to enable more control over temporal and spatial hydrogel dynamics [206]. Only locations exposed to light will react, and the extent of the reaction depends on the exposure time. There are photo-induced reactions for many bioorthogonal chemistries, termed ‘photo-click’ reactions [207], though the most notable in biomaterials literature is the thiol-ene reaction. Thiol-ene reactions occur between alkenes and thiol groups in the presence of a photoinitiator and light irradiation. The thiol groups (-SH) are initiated to thiyl radicals (-S*) following exposure to ultraviolet or visible light, depending on the photoinitiator. Irgacure 2959 is a common photoinitiator for ultraviolet activation, while Eosin Y can be used with the more cytocompatible range of visible light. Because photo-click reactions require a light stimulus of a specific wavelength, it is possible to perform multiple photo-click reactions within the same material by combining UV and visible light stimuli [208]. Additional control may be gained by adding photo-cleavage reactions to unmask or degrade chemical moieties with a different light wavelength [209,210]. One caveat is that thiol–based reactions are not perfectly biorthogonal, since thiol groups are naturally present in the amino acid cysteine. Nonetheless, these reactions often do not interfere with biological processes since cysteine is relatively low in abundance, and thiols are mostly present as disulfide bridges [207]. 

Polymers functionalized with norbornene are commonly used to enable photo-click crosslinking. For example, Jivan et al. used PEG-norbornene crosslinked with dithiothreitol (DTT) to form PEG hydrogel microparticles varying in size from 8 to 30 μm in diameter [178]. Given the potential for norbornene to also participate in IEDDA reactions, the researchers subsequently added tetrazine-functionalized proteins for conjugation with the remaining unreacted norbornene groups. Instead of using thiol groups on the crosslinker, Sawicki and Kloxin created hydrogels of thiolated PEG crosslinked with vinyl-containing peptides. The vinyl group was incorporated by functionalizing the peptides with alloxycarbonyl, a common protective group used during peptide synthesis. Using the photoinitiator lithium acylphosphinate, the hydrogels formed rapidly under cytocompatible doses of long-wavelength ultraviolet light (365 nm) [173]. Alternatively, it was recently shown that the thiyl radicals needed to initiate thiol-ene crosslinking can be generated using hydrogen peroxide in the presence of the enzyme horseradish peroxidase [211]. This approach preserves temporal control over the reaction while averting photodamage to live cells.

### 4.4. Thiol-Michael Addition

The thiol-Michael addition is a type of thiol-ene reaction that is base-catalyzed as opposed to radically mediated [212]. These reactions occur between nucleophilic thiol groups, known as the Michael donor, and electrophilic-ene compounds, or Michael acceptors, without the need for light radiation [212]. There are also Michael-type addition reactions not involving thiols, but the key attribute of the thiol-Michael addition reaction for biomaterials science is the wide array and availability of possible reactive macromers. Thiols react with several moieties commonly used in biomaterials, including maleimides, vinyl sulfones, acrylates and acrylamides, acrylonitriles, and methacrylates [177]. For example, a cyclized vinyl polymer of PEG was used as a precursor for hydrogel nanoparticle synthesis crosslinked with DTT [213]. After crosslinking, there is also potential for a retro Michael-type reaction with thiol exchange, which imparts degradability in response to reducing conditions in the microenvironment [210]. More about the thiol-Michael addition and primary uses for each reactive pairing in biomaterials can be found in [212].

Another major benefit to the thiol-Michael addition is the potential to use cysteine-containing peptides for easy and versatile biofunctionalization or crosslinking. In their study, Zhao et al. found that the inclusion of dithiol-containing peptides with laminin-derived sequences of IKVAV and YIGSR leads to a slower gelation rate but also improves the viability and metabolic activity of murine neural progenitor cells [213]. As with other examples, acryl hydrazide HA was loaded with thiol-containing RGD peptides and then crosslinked using an enzyme-cleavable di-thiol peptide [214]. Tam and colleagues leveraged thiol-maleimide click chemistry as well as biotin-streptavidin chemistry to covalently immobilize two separate bioactive factors into thiolated methylcellulose: maleimide–GRGDS for cell adhesion, and maleimide-streptavidin for subsequent binding with biotinylated platelet-derived growth factor-s (rPDGF-A), an oligodendrocyte differentiation factor [215]. 

The same group later created thiolated MC for injection into the intrathecal space on top of the spinal cord, functionalization with maleimide-modified peptide and crosslinked using PEG-bismaleimide [216]. The peptide exhibited homology to a region on an enzyme of interest, chondroitinase ABC, released over time to degrade the glial scar tissue and promote neuronal differentiation of transplanted human neural stem cells. Similarly, HA hydrogels based on thiol-alkyne Michael type addition promoted in vivo survival of encapsulated human neural progenitor cells after transplantation into a stroke cavity [214]. The HA backbone was functionalized with cell-adhesive peptides to support the survival of encapsulated neural progenitor cells as well as thiol-modified heparin, to facilitate growth factor retention. These authors used a design of experiments approach to optimize the concentration of the chosen peptides, but others have directly evaluated the landscape of integrin-binding and MMP-degradable sequences in neural tissue [217]. Proteomic analysis of the adult human brain tissue identified nine different integrin binding sites and five different MMP-cleavage sites of high interest. These sites were then incorporated into PEG-maleimide hydrogels using cysteine-containing peptides and sequential Michael-type addition reactions, thereby creating brain ECM-mimicking hydrogels that maintained astrocytes in a stellate, quiescent morphology. 

### 4.5. Click Applications for Immunomodulation

The chemo-selective nature of biorthogonal reactions makes them an ideal choice for incorporating functional groups that modulate cell-material interactions and instruct immune cell responses. For example, Glass and colleagues optimized a process to functionalize mannose onto the surface of polymeric nanoparticles using a CuAAC reaction. The conjugated sugar acted as a targeting moiety on the nanoparticle surface by promoting specific interactions with the mannose receptor CD206 that is preferentially expressed on M2 macrophages [218]. The authors demonstrated an optimal range for minimizing copper-induced cytotoxicity while maintaining high levels of conjugation efficiency and increased cell targeting. This approach enabled the successful delivery of therapeutic siRNA and subsequent reprogramming of the M2 macrophages toward an anti-cancer, M1 phenotype. A similar but opposite approach may therefore also work to redirect M1 macrophages toward M2 phenotypes to promote tissue repair. 

Bioorthogonal conjugation can add functionality or targeting moieties onto surfaces. Inert beads were chemo-selectively conjugated with a modified TGF-β1 via a Staudinger ligation between an azide and triarylphosphine [219], and the engineered surfaces efficiently converted naïve T cells to functional Treg cells for immune suppression. Dendritic cells were also labeled in vivo using azide-modified glycans [220]. The cells were recruited to the implantation site using alginate gel releasing GM-CSF as well as an azide-modified sugar, which was taken up by the dendritic cells and used to improve the generation of antigen-specific, CD8+ T cells through covalent capture of DBCO-bearing antigens and adjuvants like IL-15 and IL-15Rα. Hence, biorthogonal chemistries can advance the therapeutic efficacy of immunomodulation across a range of biomaterial-based systems. 

## 5. Click-Based Materials for Neuroimmune Modulation

Traditional immunotherapy often uses a biomaterial as drug delivery vector, missing out on the opportunity to leverage cell-material interactions as an active therapy element. Combining the efficacy of biomaterial-based immunotherapy with the versatility of bioorthogonal chemistries is an excellent strategy to create the more intentional or synergistic effects that may be required to treat complex conditions like neuroinflammation. Current literature approaches have implemented click chemistry in two primary ways: bioorthogonal crosslinking and/or selective bioconjugation. Such applications include the incorporation of cell-adhesion sites, bioactive cytokines, and degradable crosslinks toward modulating immune, glial, or neuronal cell outcomes. Nonetheless, there remains a vast unexplored space in which to leverage bioorthogonal reactions for enhancing material functionality and regulating neuroinflammatory environments. 

### 5.1. Biomaterial Scaffolds and Hydrogels

Hydrogel-based strategies have seen the highest use of bioorthogonal chemistries for directly or indirectly treating neuroinflammation. It is worth noting that each study highlighted below used a foundation of hyaluronic acid, the primary matrix component of the CNS. One notable example is Thompson and collaborators, who developed a modified HA hydrogel for treating spinal cord injuries. The authors used a mixture of furan- and methylfuran-modified high molecular weight HA (250 MDa) crosslinked with PEG-bismaleimide by a Diels-Alder reaction [198]. Prior to crosslinking the gels, the researchers physically blended in cell-derived ECM from either fibrous or protoplasmic populations of astrocytes (differentiated from RW.4 mouse embryonic stem cells). Upon implantation into the hemisected rat spinal cord, ECM from protoplasmic astrocytes decreased the size of the glial scar, decreased macrophage and microglia infiltration, and increased axonal ingrowth. Additional encapsulation of mouse embryonic stem cell-derived V2a interneurons further increased neuronal processes in and around the lesion. Hence, these bioorthogonal hydrogels enabled gentle but efficient encapsulation of ECM molecules and motoneuron cells to encourage spinal cord regeneration. 

A di-functional HA was created by modifying the primary hydroxyl and carbonyl groups of the HA backbone with thiol and azide groups, respectively [221]. The hydrogels were then crosslinked with methacrylated HA by a photo-click reaction between the thiol and methacrylate. The ratio of methacrylate HA was varied to determine the effects of difunctionalization on mouse embryonic stem cell differentiation. A 1:2 ratio for difunctional to methalcrylate HA was selected for testing in a rat model of subacute contusion spinal cord injury. In vivo gelation relied solely on the Michael-type addition reaction for crosslinking. Compared to saline controls, the 1:2 hydrogel significantly reduced infiltration of macrophages into the spinal cord lesion one week after implantation but did not significantly influence the lesion volume, astrocyte activation, or the presence of axonal staining. While not conducted in this study, the free azide functional side groups could be reacted with DBCO-modified cytokines or adhesive peptides as additional bioactive signaling molecules, which may lead to further enhancements of axonal in-growth and tissue sparing.

The introduction of macroporosity will further facilitate vascularization and immune cell modulation after injury. One such method to produce macroporous hydrogel scaffolds is by annealing together microgels [105]. Microgels are crosslinked hydrogel microparticles that can be used to encapsulate and deliver cells, proteins, and/or small molecules. One study used click chemistry to incorporate three peptides into HA-acrylamide: one peptide of cell-adhesive RGD and two others for the active clotting enzyme Factor XIIIa [203,222]. The precursor solution could be fluorescently labeled using a maleimide-containing fluorophore, and the decorated HA-acrylamide was crosslinked by a Michael-type addition with di-cysteine-containing, MMP-cleavable peptides. Finally, the microgels were linked together using Factor XIIIa, forming an annealed-solid hydrogel scaffold with macroporous void space. Applying the annealed HA microgels into a stroke cavity lowered the percentage of both astrocytes and microglia in the peri-infarct area. There was also a significantly higher percentage of vessels in the peri-infarct area and more proliferating neural progenitor cells migrating toward the damaged tissue. Collectively, the annealed HA microgels accelerated brain repair after stroke by modifying the extent of astrogliosis, inflammation, vascularization, and neural progenitor cell recruitment to the lesion. 

### 5.2. Therapeutic Particles 

The BBB is a major hurdle for delivering therapies to the CNS, even following injury or ischemia [223]. Cationic, BBB-penetrating nanoparticles comprising poly(β-amino esters) were created by reacting excessive PEG-diacrylate with 2-propynylamine via a Michael-type addition reaction [224]. The poly(β-amino esters) were then functionalized with an azide-containing nerve growth factor mimetic by azide-alkyne cycloaddition. Further, thiol-containing adhesive peptide (CYIGSR) was conjugated to the terminal acrylate groups through a thio-ene photo-click reaction. The functionalized polymeric nanoparticles had a mean diameter size of ~30 nm and promoted neurite outgrowth of cultured sensory neurons below 100 ng/mL. The nanoparticle micelles were able to passively penetrate the BBB, potentially due to the positive charge of the particles and the negative charge of the endothelial cell layer [225]. Moreover, the incorporation of YIGSR peptides positively increased the surface zeta potential, further enhancing BBB permeation. To test the effects on the neuroinflammatory response, the authors used a cryogenic lesion model of brain injury with tail vein injection of the micelle formulation every other day. The dual-functional micelles promoted decreased lesion area and BBB leakage by day 8. The treatment group also had the lowest number of GFAP^+^ cells in the injured tissue, suggesting low astrocyte reactivity.

A glycoengineering approach can be used to attach reactive groups or drug cargo onto cells for on-demand control of phenotype polarization [226]. Li and colleagues engineered cultured microglia to home to damaged vasculature by fusion with platelet-derived vesicles, and the hybrid cells were fed an azide-modified glycan that became metabolically incorporated into mucin-type glycoproteins [227]. Separately, the anti-inflammatory cytokine IL-4 was encapsulated into liposomes containing both a DBCO-PEG-modified lipid and a molecule that sensitizes the membrane to ultrasound, known as protoporphyrin IX. Excessive azide-PEG was added to quench unreacted DBCO groups and avoid liposome aggregation, and the sono-sensitive DBCO-liposomes were linked to the hybrid microglia as a composite cellular therapy. In cell culture studies, exposure to ultrasound-induced the release of the IL-4, causing upregulation of M2 markers such as Arg-1 and TGF-β. The composite therapy significantly reduced the lesion volume in a mouse model of ischemic stroke, as well as decreased astrocyte activation and increased microglia recruitment. Ultimately, triggered release of IL-4 in the stroke cavity increased the proportion of M2 phenotypes, thereby remodeling the neuroinflammatory environment to promote neuroprotection and improve functional recovery. 

While the above platelet-derived vesicles had to be generated ex vivo, extracellular vesicles can be easily isolated for tethering to biomaterials or the cell surface [228]. One study used CuAAC to conjugate cell-adhesive RGD to the exosome surface [136]. The RGD-conjugated exosomes were found to accumulate more within an ischemic brain lesion owing to the upregulation of the integrin αvβ3 in reactive cerebral vascular endothelial cells. Using the engineered exosomes to deliver curcumin, a natural anti-inflammatory agent, significantly suppressed the inflammatory response and cellular apoptosis in the lesion region compared to soluble curcumin or unloaded exosomes alone. Mice treated with the exosomes expressed significantly less TNF-α, IL-1β, and IL-6, demonstrating the ability to harness the intrinsic immunosuppressive potential of mesenchymal stem cell-derived exosomes for targeted delivery of additional anti-inflammatory agents. 

### 5.3. Hydrogel Coatings

Lastly, biomaterials can be used as coatings to create a new, amenable cell-material interface. A major hurdle for sustained use and recording from neural electrodes is that the required stiff materials cause extensive inflammatory scarring. Matching the mechanical properties of an electrode to that of brain tissue can minimize the scarring response [229]. The unique qualities of thiol-ene photo-click chemistry offer precise control over the mechanical properties and thermal glass-transition temperature of a polymer, making it an ideal electrode coating. By varying the monomer composition, a thiol-ene shape memory polymer coating can be designed to be stiffer at room temperature for tissue insertion but softer once equilibrated with brain tissue. Dip coating a uniform layer of thiol-ene shape memory polymer onto silicon microelectrode reduced astrocytic scarring after 16 weeks [230]. Using the thiol-ene polymer coating to reduce the mechanical mismatch between the device and tissue was comparable to, if not better than, the bare silicon microelectrodes. While the recording efficacy needs to be evaluated, this work suggests bioorthogonal biomaterials could lengthen the allowable time for intracortical measurements. 

Bioorthogonal coatings can also be used to encapsulate single cells in a hydrogel casing for enhanced cellular protection and control of the microenvironment [231]. Azide-modified sugars were metabolically incorporated onto the surface of neural stem cells in culture for subsequent reaction with DBCO-PEG chains of various molecular weights. The polymeric coating improved cell survival and modulated growth factor release. Varying the PEG molecular weight also enabled control over the stiffness of the immediate pericellular environment. Specifically, higher molecular weight PEG (20–30 kDa) produced a softer polymeric ECM and significantly increased the release of neurotrophic factors such as VEGF and CNTF. Similar single-cell hydrogel approaches may therefore be useful for protecting transplanted cells from hostile or mechanically stiff environments, favoring neuronal differentiation over astrocytic differentiation, or sustaining an anti-inflammatory phenotype in pre-stimulated cellular therapies [143,148,232]. 

## 6. Conclusions and Future Directions

Pathological inflammation of the CNS is complex and difficult to regulate, but the versatility and effectiveness of biomaterial-based strategies suggest this will become the next frontier of neuro-immunomodulation. Therapeutic material approaches will benefit from the specificity and utility afforded by bioorthogonal or ‘click’ chemistry. A majority of previous bioorthogonal materials applied to the CNS have assessed neuroinflammation as a secondary outcome opposed to an intentional target, creating vast opportunities for further impactful advancements. Exciting future directions not yet applied to neuroinflammation include pre-targeted imaging, personalized or modular drug delivery, theranostics, and temporally dynamic therapies. Consider bioorthogonal chemistry as the perfect toolbox for realizing dynamic materials that may one day enable resolving the chronic glial scar, recapitulating an M1 → M2 transition of successful wound healing, or reciprocally interacting and guiding the CNS toward regeneration. 

## Figures and Tables

**Figure 1 ijms-23-08496-f001:**
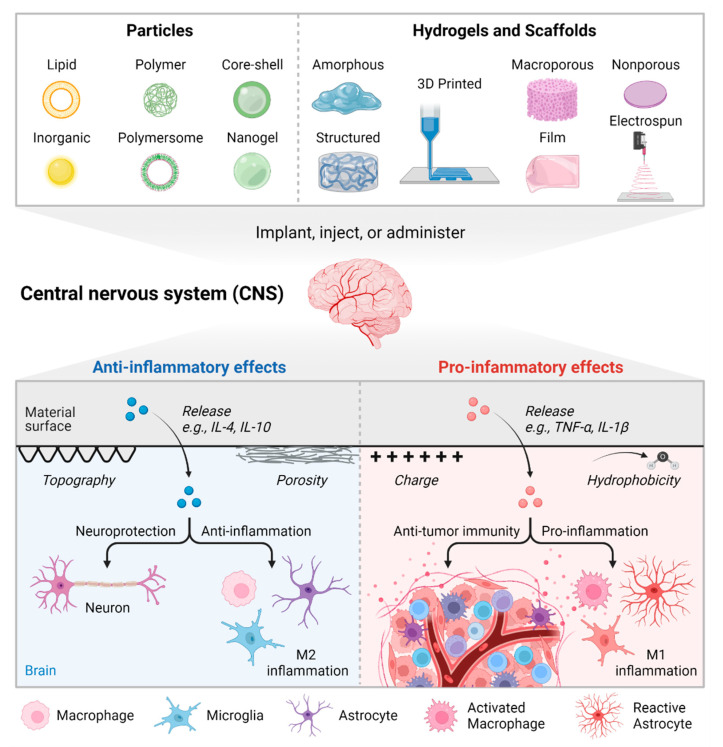
Overview of engineering material properties to instruct inflammatory outcomes. Created with Biorender.com and exported under a paid license.

**Figure 2 ijms-23-08496-f002:**
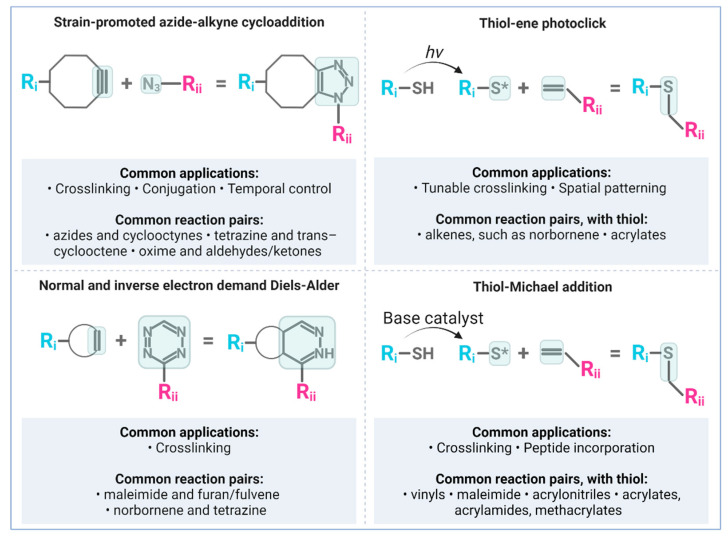
Bioorthogonal reaction schemes commonly used in biomaterials, including their respective applications and reactive functional groups. Created with Biorender.

## Data Availability

Not applicable.

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
