# Peer review of "Clickable Biomaterials for Modulating Neuroinflammation"

_ijms, 2022, doi:10.3390/ijms23158496_

Round 1

Reviewer 1 Report

I have carefully read the review by Cornelison and Fadel entitled "Clickable biomaterials for modulating neuroinflammation". The manuscript is well written, organized and is a significant contribution to this emerging field of reserach. I have only few issues to raise:

-line 146 "Infiltrating macrophages and resident microglia acutely exhibit an anti-inflammatory phase that transitions into a prolonged, persistent pro-inflammatory phase". It would be interesting a comment by the authors to specify duration of acute anti-inflammatory phase versus persistent pro-inflammatory phase.

-through the text the adverb "recently" is not always used appropriately, (as an example line 298 ref 144)

-line 309 " an pro-inflammatory" please correct

Author Response

Thank you for your kind comments and suggested edits. We reevaluated the use of "recently" and removed the word on several occasions. We also corrected the term "pro-inflammatory" and added a statement regarding the transition and timeline of anti- and pro-inflammatory phases. We feel these changes have improved the messaging of our manuscript, so thank you again for your valuable input. 

Reviewer 2 Report

In " Clickable biomaterials for modulating neuroinflammation", the authors have reviewed "recent advances in the application of click-based biomaterials for treating neuroinflammation and promoting neural tissue repair." The good general introduction reviews pathologies involving neuroinflammation with sufficient detail and good referencing for readers to do deeper searches where needed in to neurological disease pathology that would interest that audience. The discussion of microglia and astrocytes is a welcome addition in defining the crosstalk between immigrating leukocytes and resident immune cells. The description of the balance between pro-inflammation and resolving states of cell signaling and the importance of timing and amplitude of that biology is important. This unbiased, balanced consideration of pros and cons continues in the description of current materials and continues throughout the review. The number of references seems large, however, for a review of this nature that integrates discussions of targeted pathology, immunology, chemistry, biomaterials and applications, the reference number is appropriate and useful to the reader.  Overall the review is well done.

Major concerns:  None.

Minor concerns or opportunities:

1. This is a well planned and executed review. Think about possible audiences. That is, for many people in neurological diseases, immunology, and some in regenerative medicine, the biology is the goal.  Thus the chemistry in this review, though excellent, is a tool to be considered and used. Consequently, those readers would benefit from a few schematic examples to accompany each reaction chemistry in section 4, a chance for them to learn the click chemistry and begin to perhaps integrate the biology and biochemistry they know with the click chemistry that they do not. The experienced click chemist will be fine, and has ample references to seek out the biology when needed.

2. There are a few typos/verb tense errors etc. For example, line16 of the abstract should be "interfere".  Line 196 should be cells "are". There are others to be fixed when combing through the final copy. Lines 437-438 are bold face fonts?

Author Response

Thank you for your time and valuable input. We agree graphical representations will help a broader audience to better grasp the contents of the review. We have therefore added two figures: one to visually summarize the different materials and applications in the central nervous system, and one figure to depict the specific chemistry mentioned. We have also thoroughly reviewed and edited the manuscript for typographical errors. We feel the manuscript now better conveys the message and is understandable to a larger audience, so thank you again for your thoughtful suggestions.